# Factors Influencing Driving following DBS Surgery in Parkinson’s Disease: A Single UK Centre Experience and Review of the Literature

**DOI:** 10.3390/jcm12010166

**Published:** 2022-12-25

**Authors:** Luciano Furlanetti, Asfand Baig Mirza, Ahmed Raslan, Maria Alexandra Velicu, Charlotte Burford, Melika Akhbari, Elaine German, Romi Saha, Michael Samuel, Keyoumars Ashkan

**Affiliations:** 1Institute of Psychiatry, Psychology and Neuroscience, King’s College London, London SE5 8AB, UK; 2King’s Health Partners Academic Health Sciences Centre, London SE1 9RT, UK; 3Department of Neurosurgery, King’s College Hospital NHS Foundation Trust, London SE5 9RS, UK; 4Department of Neuropsychology, King’s College London, London SE5 8AB, UK; 5Department of Neurology, King’s College Hospital NHS Foundation Trust, London SE5 9RS, UK

**Keywords:** deep brain stimulation, Parkinson’s disease, driving, subthalamic nucleus, rehabilitation, neurodegenerative diseases, ageing

## Abstract

Parkinson’s disease (PD) is a complex neurodegenerative disorder, leading to impairment of various neurological faculties, including motor, planning, cognitivity, and executive functions. Motor- and non-motor symptoms of the disease may intensify a patient’s restrictions to performing usual tasks of daily living, including driving. Deep Brain Stimulation (DBS) associated with optimized clinical treatment has been shown to improve quality of life, motor, and non-motor symptoms in PD. In most countries, there are no specific guidelines concerning minimum safety requirements and the timing of return to driving following DBS, leaving to the medical staff of individual DBS centres the responsibility to draw recommendations individually regarding patients’ ability to drive after surgery. The aim of this study was to evaluate factors that might influence the ability to drive following DBS in the management of PD. A total of 125 patients were included. Clinical, epidemiological, neuropsychological, and surgical factors were evaluated. The mean follow-up time was 129.9 months. DBS improved motor and non-motor symptoms of PD. However, in general, patients were 2.8-fold less likely to drive in the postoperative period than prior to surgery. Among the PD characteristics, patients with the akinetic subtype presented a higher risk to lose their driving licence postoperatively. Furthermore, the presence of an abnormal postoperative neuropsychological evaluation was also associated with driving restriction following surgery. Our data indicate that restriction to drive following surgery seems to be multifactorial rather than a direct consequence of DBS itself. Our study sheds light on the urgent need for a standardised multidisciplinary postoperative evaluation to assess patients’ ability to drive following DBS.

## 1. Introduction

Deep Brain Stimulation (DBS) has become a standard European approach and FDA-approved treatment for patients diagnosed with a range of neurological and psychiatric conditions refractory to medical management, such as in selected cases of Parkinson’s disease (PD), dystonia, essential tremor, obsessive compulsive disorder, and epilepsy [1,2]. The field of neuromodulation is one of the fastest-growing neuroscientific areas. Over the last decades, DBS technology has improved and research in neuromodulation has expanded exponentially, allowing for the development of promising DBS treatments for several neuropsychiatric conditions beyond the field of movement disorders [1,3,4]. Despite the large worldwide experience with DBS, especially in the management of PD, there is only limited evidence in the literature concerning the impact of the stereotactic implantation of electrodes in deep brain structures for chronic neuromodulation on patients’ ability to drive [5,6]. PD is a complex neurodegenerative disorder, leading to impairment of various neurological faculties, including motor, planning, cognitivity, and executive functions [7,8]. Motor- and non-motor symptoms of the disease may intensify a patient’s restrictions to performing usual tasks of daily living, including driving [9,10,11,12,13]. Several works have consistently shown a positive impact of DBS on motor and non-motor symptoms of PD, and consequently, on patients’ overall health-related quality of life (HRQoL) [14,15,16,17,18]. Although the ability to drive directly impacts a patient’s independency to carry on with previously performed daily activities, driving is often overlooked as an indicator of quality of life in PD studies [10,19].

PD patients often have impaired driving ability due to a combination of motor symptoms, as well as cognitive, emotional, and visual impairments [10]. Motor and non-motor manifestations of the disease refractory to optimal medical management, associated with motor complications, unpredictable on-off phenomenon, and common side-effects of antiparkinsonian medications, such as daytime sleepiness and intrusive dyskinesia, may increase the risk of road traffic accidents in this population, even in early stages of the disease [9,10,12,19]. Indeed, in the UK, patients have a legal obligation to inform the UK driving authority upon diagnosis of PD, and to also inform specifically should drug side-effects that could impair driving develop. Further, notification may also be needed after DBS insertion. Most PD patients, mainly those with controlled symptoms without troublesome fluctuations and without significant sleep, psychiatric, cognitive, or impulsive symptoms, can keep driving upon notifying their local driving regulatory authority, although they remain required to have regular medical updates as the disease progresses. Meindorfner et al. 2005 reported that 82% of 6620 evaluated PD patients held a driving license, and 60% of them were still driving [20]. Furthermore, the authors found an increased risk of causing accidents among patients feeling subjectively more impaired by the disease or presenting with daytime sleepiness and sudden onset of sleep while driving [20].

Other groups have investigated the association of neurological and neuropsychological evaluations with fitness to drive in PD [9,11,19,21,22,23,24,25,26,27,28,29,30,31,32]. However, there is currently still no consensus on which tests to use to determine driving fitness in people with PD [5,10,21], and there are different driving regulations for cars and motorcycles compared with buses and lorries. According to the literature, the predictive accuracy of the available screening tools ranges between 72% and 90% in parkinsonian subjects [21,25,30]. Furthermore, only a few studies have included patients submitted to surgical management with DBS [5,6,33]. Thus, driving authorities often require regular written confirmation of fitness to drive from the medical team to renew driving permits [34]. The paucity of studies investigating pre- and postoperative clinical factors influencing a patient’s ability to drive following DBS surgery may lead DBS teams to arbitrarily define individual recommendations based on their own experience and expose themselves to potential medicolegal issues in case of driving accidents involving operated PD patients.

In the present study, epidemiological, neuropsychological, clinical, and surgical factors influencing the ability to drive in PD patients submitted to DBS were investigated and discussed in the light of the current literature.

## 2. Materials and Methods

### 2.1. Study Design, Study Groups, and Data Collection

Medical records of 125 consecutive patients who underwent deep brain stimulation (DBS) surgery for the management of Parkinson disease (PD) were reviewed. All patients were assessed pre- and postoperatively by a multidisciplinary team, at a single institution, between 2002 and 2018. Routine preoperative evaluation included detailed neurological, neuropsychological, psychiatric, and neuroimaging review prior to discussion by a multidisciplinary team. Clinical, epidemiological, neuropsychological, and surgical factors that could influence the ability of the patients to drive following DBS surgery were evaluated. The inclusion criteria were the clinical diagnosis of idiopathic Parkinson’s disease for at least 5 years, levodopa-responsiveness, and refractory parkinsonian symptoms, motor fluctuations or dyskinesia despite optimisation of medical management [35]. Exclusion criteria were dementia, untreated psychiatric disorders, or the presence of any clinical or surgical contraindications. Clinical data were obtained during routine clinical care.

### 2.2. Imaging, Stereotactic Planning and Surgical Procedure

Standard planning MRI sequences were obtained using a 1.5 T General Electric (GE) MRI scanner (GE Healthcare, Chicago, IL, USA) or a 1.5 T Siemens MRI scanner (Siemens, Erlangen, Germany) either preoperatively up to two weeks before surgery or stereotactically on the day of surgery. MRI sequences consisted of volumetric T1-weighted and 2 mm thick contiguous through target T2-weighted axial slices for the subthalamic nucleus (STN) and thalamic targets. For the globus pallidus internus (GPi) target, a proton density-weighted scan was performed using a repetition time (TR) of 5630 ms, an echo time (TE) of 15 ms, a slice thickness of 2 mm, field of view (FoV) of 250 mm, flip angle of 250 degrees, base resolution of 256 mm, and a voxel size of 0.5 × 0.5 × 2 mm^3^ [36]. Stereotactic planning was performed using a Surgical Navigation System (Medtronic, Minneapolis, MN, USA). Surgical procedures were performed using a Leksell G frame (Elekta, Stockholm, Sweden) under local or general anesthesia according to the conditions and needs of the patients. All patients had a preoperative stereotactic imaging with MRI, CT, or the O-arm, as well as an immediate postoperative stereotactic verification scan [37]. Additional intraoperative scans for verification of the DBS lead were performed when necessary. Direct targeting was used for stereotactic planning when implanting the STN or GPi. Atlas coordinates were used for the ventral intermediate nucleus of the thalamus (VIM) [38]. Standard stereotactic implantation of the DBS electrodes was performed in two stages: insertion of the intracranial leads in the first stage was followed by insertion of the extensions and battery in the second stage of the procedure. Surgical planning and procedures were performed or assisted by a senior neurosurgeon for all study participants. Intraoperative microelectrode recording was not used in this cohort. All patients were reviewed monthly for 3 postoperative months, and every 3 months thereafter. The monopolar review was usually performed at 4 weeks postoperatively and stimulation started, using standard parameters, e.g., 60 µs at 130 Hz, with amplitude and medications further adjusted individually according to the patient’s clinical response during the follow-up period.

### 2.3. Outcome Assessment

Patients undergoing deep brain stimulation for the management of PD were routinely reviewed pre- and postoperatively by a multidisciplinary team. Several neurological and neuropsychological parameters were investigated as potential predictors associated with the ability to drive pre- and/or postoperatively. Data regarding epidemiological factors, such as gender, age of onset of PD symptoms, age at surgery, timespan from first diagnosis of PD to surgery, levodopa equivalent daily dose (LEDD), Hoehn and Yahr classification, surgical aspects, i.e., brain target, postoperative complications, and extensive neuropsychological evaluation, addressing mood, intellectual function, memory, language, perceptual, cognitive, and executive functions were systematically and prospectively collected during routine pre- and postoperative multidisciplinary assessments.

#### Neuropsychological Evaluation

Neuropsychological evaluations were performed prior to surgery, with the patients on antiparkinsonian medications, and within the first year postoperatively, with the patients on medication/ on stimulation. The neuropsychological assessments included the Wechsler Adult Intelligence Scale (WAIS-III), the Hayling Sentence Completion Test, the Brixton Spatial Anticipation Test, the Visual Object and Space Perception Battery (VOSP), the Camden Short Recognition Memory Tests, the Trail Making Test from the Delis–Kaplan Executive Function System (D-KEFS), and the Hospital Anxiety and Depression Scale (HADS). The WAIS-III is a widely used IQ test designed to measure general intellectual functioning in adults and adolescents. The WAIS-III consists of a subsequent revision of the WAIS test, providing scores for verbal, performance, and full-scale IQ, along with secondary indices for verbal comprehension, working memory, perceptual organization, and processing speed. The Hayling Sentence Completion Test is a measure of response initiation and response suppression, consisting of two sets of 15 sentences, each having the last word missing [39]. The test is entirely spoken and therefore suitable for people with difficulties involving reading, visual perception, or movement. The Brixton Test is a visuospatial sequencing task with rule changes [39]. The test measures the ability to detect rules in sequences of stimuli. Both tests are used to assess executive functioning in patients with neurological disorders, including PD [39]. The VOSP assessment consists of eight subcomponent tests for evaluating object and space perceptual disturbances, such as incomplete letters, object decision, position discrimination, number location and cube analysis [40]. The Trail Making test was developed to isolate set-shifting from other component skills, such as letter sequencing and visual scanning, and has been shown to detect executive functioning impairment in a variety of neurological disorders [41]. Clinical and neuropsychological baseline and postoperative parameters were acquired prospectively and by the same team.

### 2.4. Review of the Literature

The databases PubMed, Medline, EMBASE, and Scopus were used to conduct searches. The Medical Subject Headings (quality of life) AND (Parkinson’s disease) AND (ability to drive) AND (deep brain stimulation) were used for searches. Results were limited to those published in English, from 2000 to 2019. The inclusion criteria were: (1) to be a primary peer-reviewed clinical study, (2) to have included patients with diagnosed Parkinson’s disease, (3) to have reported data on the impact of clinical and/or surgical management of PD on quality of life and patient’s ability to drive, age group and number of patients studied, and modality of clinical/surgical treatments performed. Studies not published in full-text or containing incomplete information were excluded.

### 2.5. Statistical Analysis

Data were presented as the mean ± standard error of the mean (SEM) and as percentages. Data were entered onto a Microsoft Excel worksheet, and statistical analysis was performed using SPSS version 23 software (IBM Corp., Armonk, NY, USA). A paired sample *t*-test was used to compare variables with a normal distribution and the Wilcoxon signed-rank nonparametric test was used for variables that did not follow a normal distribution. Patient characteristics such as age and profession, surgical characteristics such as DBS target, the presence and type of complications, and the need for revision operations; PD characteristics like hand dominance, PD subtype, laterality at PD onset, Hoehn and Yahr preoperative score, LEDD, as well as the presence of neuropsychological abnormalities and their type; and the results of several psychological tests were analysed as potential risk factors for surgical complications using logistic regression. All the factors included in the logistic regression analysis were transformed into categorical variables. The results of the psychological tests were dichotomised into improved/worsened according to their changes between the pre- and postoperative period. The potential risk variables were assessed against the outcome considered as the ability to drive post-operatively. The sample size requested to support the logistic regression was calculated using the formula N = 10 k/p (p-smallest proportion of negative or positive cases, k-number of independent variables, N-minimum number of cases to include). According to the minimum estimated value, the sample size was considered adequate to support the logistic regression. To avoid overfitting of the model, a stepwise approach was used for each variable and only the statistically significant variables were retained. Our sample size can support a logistic regression with three independent variables without overfitting of the model. Missing values or incomplete data have been excluded from the analysis. A *p*-value <0.05 was considered statistically significant.

## 3. Results

A total of 125 consecutive patients treated with DBS for the management of Parkinson’s disease were included; 68.8% (*n* = 86) were male. The mean age of patients at surgery was 63.5 years (range 38–82) and 49.6% of them (*n* = 62) had at least two other known comorbidities. The mean timespan from the first diagnosis of PD to surgery was 10.5 years (range 5–30) and the mean follow-up time until last outpatient consultation was 129.9 months (range 49–240). The epidemiological features of the cohort are summarized in Table 1.

### 3.1. Preoperative Clinical Assessment

At the time of preoperative evaluation, 77 out of 125 patients (61.6%) were still able to drive. Fourty-four percent (*n* = 55) were retired and thirty-two percent were formally working. Eighty-six percent (*n* = 108) of the patients were right-handed; the laterality onset of PD symptoms was evenly distributed and did not correlate with hand dominance. Most patients (86.6%) ranged between modified Hoehn and Yahr classification 2 and 3, and presented the akinetic-rigid subtype of PD, whereas 48.8% (*n* = 61) of the patients had intrusive resting or postural tremor associated. The mean preoperative levodopa equivalent daily dose (LEDD) at the time of preoperative assessment was 1096 mg (range 225–2980 mg), and the mean percentage improvement on the UPDRS part III scale on levodopa challenge was 56.1% (range 20–90%). The Subthalamic nucleus (STN), the Ventral intermediate nucleus of the thalamus (VIM) and the Globus pallidus internus (GPi) were chosen as targets for DBS in 82.4% (*n* = 103), 9.6% (*n* = 12) and 8.0% (*n* = 10) of the patients, respectively. All patients underwent a through preoperative neuropsychological evaluation, as previously described. Thirty-eight percent of the patients were identified with some type of abnormality, e.g., mood, cognitive disorders, impulse-control behaviours (ICB), on the neuropsychological tests, which had to be addressed before surgical intervention. Twenty-three out of one hundred twenty-five patients (18.4%) were under treatment for mood disorders, such as anxiety and depression, 14.4% had ICB and 9.6% had mild cognitive symptoms.

### 3.2. Postoperative Clinical Assessment

Fourty-seven percent of the patients (*n* = 59) presented unremarkable neuropsychological evaluation postoperatively. Interestingly, a 72.2% decrease of the incidence of ICB was noted, likely associated with a lower dose of medication postoperatively. At the last follow-up consultation, the mean overall LEDD across patients was 845.2 mg/d (range 0–2940 mg), which represents a mean reduction of 11.4 *±* 58.0% in comparison to prior to DBS, a relevant positive impact of surgery taking into account the inclusion of GPi- and VIM-DBS patients, and also the long-term follow-up of this cohort (range 40–240 months). On a group level, the incidence of mood disorders decreased by ca. 40% in the postoperative period. However, patients’ complaints regarding subjective mild cognitive decline postoperatively were twice as common as prior to surgery. As previously reported in several studies [42,43], a statistically significant decline in verbal fluency and performance on the WAIS-III was noted (*p* = 0.02 and *p* = 0.00, respectively). Pre- and postoperative neuropsychological outcomes are summarized in Table 2. Cronbach’s alpha reliability for the neuropsychological assessments was 0.85.

Potential factors that could be associated with the ability to drive were analyzed (Table 3 and Table 4). In general, patients were 2.8-fold less likely to drive in the postoperative follow-up than prior to surgery. In this cohort, only 27 out of 125 (21.6%) patients reported to drive postoperatively, whereas sixty-one percent (*n* = 77) of the patients were reportedly competently able to drive prior to surgery. Interestingly, however, 13% (*n* = 2) of the patients who were not fit to drive prior to DBS recovered their driving skills after surgery. The univariate analysis showed that, among the demographic characteristics, patients who were retired pre-operatively were more likely to stop driving post-operatively (OR 0.359, 95% CI [0.120, 1.074], *p* = 0.067) (Table 3). None of the surgery-related factors such as the DBS target, presenting post-operative complications or revision operations were statistically significant in the univariate regression. Among the PD characteristics, patients with the akinetic subtype presented a higher risk to lose their driving licence postoperatively (OR 2.967, 95% CI [1.071, 8.218], *p* = 0.036). Another factor to account negatively on the outcome was the presence of an abnormal neuropsychological evaluation postoperatively (OR 3.220, 95% CI [1.079, 9.604], *p* = 0.036) and irrespective of their type (OR 0.767, 95% CI [0.526, 1.120], *p* = 0.169). Although the changes between pre- and postoperative LEDD were significant, this was not a significant factor influencing the driving ability in our cohort (OR 0.584, 95% CI [0.201, 1.698], *p* = 0.323). Of all psychological tests included in the analysis, only the Hayling test results retained borderline statistical significance, showing negative outcomes among patients with higher postoperative scores (OR 9.846, 95% CI [1.003, 96.664], *p* = 0.050, Table 4). The analysis also showed a trend towards improved outcomes among patients with higher Brixton test scores, although it was not statistically significant (OR 3.556, 95% CI [0.692, 18.282], *p* = 0.129). As expected, patients with an abnormal preoperative neuropsychological test were 1.5-fold more likely to have already retired from work prior to surgery and 2.6-fold more likely to have an abnormal postoperative neuropsychological evaluation than their counterparts with unremarkable neuropsychological findings, *p* = 0.03 and *p* = 0.00, respectively. None of the variables identified in the univariate analysis remained statistically significant on the multivariate regression.

The overall complication rate in this cohort during the entire follow-up time was 29.6%: 18.4% of the patients presented transient postoperative neurological side-effects, such as confusion, dishibition, paresthesias, and speech issues, 10.4% had wound healing problems/seromas and/or infection, and 3.2% experienced hardware issues. Transient complications resolved within the first few weeks postoperatively, and were either directly related to the surgical procedure itself or to stimulation during the titration phase.

## 4. Discussion

Driving a car is an inherent part of the contemporary way of living, being considered one of the most important activities of daily life, not only for the economically active population, but also for patients relying on transportation for social independence [10,26,44]. This discussion will focus on car driving, as driving buses or lorries for employment is beyond the scope of this manuscript. Cessation of car driving can have a negative impact on quality of life, compromising patients’ independence and leading to isolation. This prospective clinical study aimed to investigate neurological and neuropsychological parameters as potential predictors associated with the ability to drive in a cohort of PD patients undergoing DBS. The findings showed that most patients were already retired from work but still driving at the time of the preoperative evaluation. In general, patients were 2.8-fold less likely to drive in the postoperative follow-up than prior to surgery. Nevertheless, the explanation for that seems to be multifactorial rather than a direct consequence of DBS itself. For instance, PD akinetic subtype, working status at the time of surgery, and the presence of an abnormal postoperative neuropsychological evaluation were all associated with driving restriction following surgery. On the other hand, patients with tremor-dominant PD and those with unremarkable postoperative neuropsychological evaluation were more likely to resume driving after surgery. Furthermore, neurological and neuropsychological assessments revealed that despite a marginal decline in verbal fluency and performance on WAIS-IIII, DBS surgery improved motor and non-motor functions, as depicted by the significant postoperative reduction of LEDD and by the reduction of the incidence of ICB and mood disorders in this cohort. Interestingly, 13% patients who were not able to drive prior to DBS due to the severity of their PD symptoms resumed driving after surgery.

PD is a complex progressive neurodegenerative disorder, leading to impairment of multiple motor, planning, cognitivity, and executive functions, as well as sleep [7,35]. On top of these symptoms, progression of motor- and non-motor complications, more common at later stages of the disease, may increase patients’ restrictions to perform usual tasks of daily living [8,45]. Notwithstanding, these symptoms, including cognitive deficits, mainly in the domains of executive function and memory, may also be present in early stages of PD [9]. Driving is a highly complex task and relies on preserved memory, cognitive, executive, sleep, and visual functions [6,10,12], which are dynamic and prone to deteriorate over time as a result of aging itself, and intensified in the context of progression of as PD. Symptoms such as intrusive tremor and bradykinesia, cognitive and emotional impairments, and visual-perceptual deficits, aggravated by unpredictable wearing-off, dyskinesia, and side-effects of medications, can lead to severe restriction of the ability to drive and increased risk of road traffic accidents [10,21,30].

Considering the ageing population and the consequent increasing incidence of neurodegenerative disorders globally, the evaluation of driving competency in this group has become a public health concern [11,19,46]. Therefore, several works have addressed the impact of aging and neurodegenerative disorders on the ability to drive [10,12,20,30,47]. Overall, in matched case-control studies assessing PD patients’ fitness to drive, PD drivers perform worse on tests of brake response and reaction, cognition, sleepiness, and contrast sensitivity, which altogether may lead to unsafe driving [10,11,21,48]. Uc et al. 2007 showed in a prospective study that the cumulative incidence of driving cessation at 2-year-follow-up was 17.6% (range 11.5–26.5%) among PD patients, being associated with older age, low driving exposure, impairment in visual perception and cognitive abilities, severity of parkinsonism, and total LEDD [48]. Thompson et al. 2018 reported a meta-analysis of 50 studies, including 5410 participants, comparing patients with PD and healthy controls on realistic on-the-road and simulator driving situations. Interestingly, PD patients presented a 6.16-fold increased risk of on-the-road failure and 2.63-fold higher risk of simulator crashes than healthy controls, although no evidence of increased risk of real-life road-traffic accidents as assessed by self-report was observed. This discrepancy could be explained in three ways. First, patients are likely to adopt compensatory and protective strategies, minimizing challenges in real-life driving. Second, there may exist a degree of bias introduced in self-reporting, due to difficulties in recalling minor incidents. Last, it may be possible that deficits observed in neurological and neuropsychological evaluations do not necessarily translate into an increased risk of road-traffic accidents [6,10,30,46]. Different from previous reports, the meta-analysis did not reveal any impact of age, sex, previous driving experience, severity of PD, disease duration, or LEDD on driving outcomes [46], suggesting that impairment of the ability to drive occurs even at younger ages and earlier stages of the disease.

In the last decades, DBS associated with optimized medical treatment and intensive physical and cognitive rehabilitation has been consistently shown to improve quality of life, motor functions, and non-motor functions in patients suffering from PD [14,16,17,45]. Several landmark studies have shown that DBS is superior to optimized medication management alone both in younger patients with early motor fluctuations and in patients with advanced PD [14,15]. Furthermore, available evidence indicates that DBS improves non-motor symptoms (NMS) such as sleep, fatigue, perceptual issues, urinary, and cardiovascular [16,17,49]. Despite that, there is no consensus on specific recommendations regarding driving following DBS in PD.

In most countries, the license holder is advised to self-report to the responsible driving agency in case of any concerns regarding their fitness to drive following surgery or in the presence of injury or illness. This manuscript cannot address worldwide differences in requirements for notification, but it should be noted that in some, e.g., the UK, the license holder is required to report a diagnosis of PD even if the driver does not feel that there is an impact of driving. In cases of cranial surgery, not particularly DBS, recommendations for restriction to drive vary largely depending on several factors, such as underlying pathology, seizure freedom and residual neurological deficits, generally ranging from 3 to 12 months [5,33,34]. In the United Kingdom, license holders have a legal duty to notify the Driver & Vehicle Licensing Agency (DVLA) of any illness that would impact their safe driving ability [34]. Specifically regarding DBS surgery, provided that the patient is seizure-free, that there is no perioperative complication from the surgery nor debarring residual impairments, according to the DVLA, driving could be resumed; the timing of return, however, is evaluated case by case [34]. Where uncertainty exists, the driver can have a formal assessment by the driving authority. Charmley et al. 2021 reported the currently used driving restrictions in Australia based on a survey of neurologists and neurosurgeons performing DBS across the country. According to the responders, the recommendations for private license-holders undergoing uncomplicated DBS surgery ranged from 4 weeks to 3 months driving restriction, although no specific guidelines on how to assess fitness to drive were available [5]. The authors proposed a 6-week driving restriction for private license holders and a 6-month driving restriction for commercial drivers following uncomplicated DBS. Additionally, commercial drivers were advised to be submitted to detailed multidisciplinary, including neuropsychological and occupational therapy assessments, prior to resuming driving [5].

Regarding the impact of DBS on PD patients’ ability to drive, Buhmann et al. 2013 showed in a prospective clinical study using a simulator of on-road driving that DBS of the subthalamic nucleus (STN) actually improved their performance [6]. Although operated PD drivers were more cautious and slower than normal controls and non-operated patients during the driving tasks, they drove more safely and committed less errors than their counterparts [6]. Furthermore, the data indicated that DBS patients did not perform worse in comparison to non-operated PD patients regarding driving. Age and cognitive deficits impacted driving performance negatively in both operated and non-operated PD groups. The authors concluded that DBS might positively impact the ability to drive, not only due to improvement of motor symptoms, but also to driving-relevant non-motor skills, such as implicit procedural learning, goal-directed action selection, and decision learning [6,50].

The influence of non-motor symptoms, such as cognitive and visual dysfunction, on the ability to drive is well recognized. Several works have investigated the ideal battery of tests that could more precisely predict fitness to drive in PD, nevertheless, there is still no consensus. Furthermore, concerning surgically managed PD patients, the experience with evaluating post-operative fitness to drive is even more sparsely documented in the literature. In our study, an abnormal postoperative neuropsychological evaluation, e.g., the presence of mood disorders, ICB, or any degree of cognitive impairment, was inversely correlated with the ability to drive post-surgery. Furthermore, a poor performance on cognitive initiation and inhibition assessed with the Hayling test was associated with restriction to drive following surgery. According to previous studies, neuropsychological evaluation addressing performance on visual attention, visuoconstructional abilities, and motion perception, as well as measures requiring rapid responding, visual spatial cognition, and executive functioning have been documented to be useful to detect unsafe driving skills in the PD population [22,24,51]. Therefore, neuropsychological assessments should include evaluation of attention, concentration, visual and verbal memory, visual perception, organization, executive function, reaction times, information processing, and cognitive flexibility [19,23,25,31,32,48]. Other than that, the presence of mood disorders, daytime sleepiness, and ICB should be regularly investigated [10,19,21]. The combination of neuropsychological screening tests and neurological evaluation may improve the prediction of patients’ fitness to drive from about 70 to 90% [25]; however, no evidence-based guidelines are available for assessing driving ability in this group of patients, indicating a need for larger clinical studies using both on-the-road and driving simulators.

Although advancements in technology, surgical skills, and increasing experience have allowed DBS to become a safe procedure, the incidence of adverse events ranges from 4.2% to 37% across studies [5,52]. The overall complication rate in this cohort was 29.6%, where the majority consisted of transient postoperative issues, such as stimulation related side-effects during the titration phase, which subsequently resolved. A proportion of patients, however, had wound/hardware-related complications at some point during the study period. The occurrence of perioperative complications may lead to prolonged hospital stays, a need for further interventions, and longer length of time for neurological rehabilitation, which may lead to prolonged restriction to drive postoperatively. Therefore, every effort must be directed towards safe surgery and perioperative management to minimize the incidence of adverse events.

Despite the prospective data acquisition, this study has limitations. The analyses were performed by the same team managing the patients, and due to the long-term follow-up, the databank was missing information for some of the patients, which could have introduced an element of bias into the results. Another important factor was the lack of a matched-control group of non-operated patients on best medical management to better evaluate the impact of surgery and of disease progression on patients’ ability to drive in the long term. Additionally, due to the heterogeneity of the psychological tests performed, the sample size for a given individual test was more limited. Thus, the results of the tests were dichotomised into worsened or improved to account for the reduced sample size, which would perhaps fail to capture subtle but significant changes among the categories analysed. Therefore, the present findings would need further investigation in a multicenter study setting, which would allow long-term evaluation of a larger cohort of patients.

## 5. Conclusions

There is a limited literature exploring the impact of DBS on patients’ ability to drive following surgery. The present data reveal that restriction to drive following surgery seems to be multifactorial rather than a direct consequence of DBS itself, and reinforce the importance of routine postoperative neuropsychological evaluation in PD. Larger multicentre studies using both on-road driving assessments and driving simulators need to be carried out in combination with neurological and neuropsychological evaluations to better understand predictors of postoperative safe driving. Our study sheds light on the urgent need for a standardised multidisciplinary postoperative evaluation to assess patients’ ability to drive after DBS. This will provide a framework to which patients, medical teams, and driving authorities will be able to refer in evaluating fitness to drive and enabling enhanced quality of life outcomes where possible.

## Figures and Tables

**Table 1 jcm-12-00166-t001:** Epidemiology.

	*n*	%
Age (mean)	63.5 ± 8.4	-
Gender		
Male	86	68.8
Female	39	31.2
Hand dominance		
Right-handed	108	86.4
Left-handed	9	7.2
Onset of PD symptoms		
Right	49	39.2
Left	43	34.4
Bilateral	7	5.6
N.A.	26	20.8
PD subtype		
Akinetic	94	75.2
Tremor-dominant	23	18.4
Mixed	8	6.4
H&Y at surgery (mean)	2.1 ± 0.5	-
Timespan Diagnosis to Surgery (years)	10.4 ± 4.7	-
Comorbidities		
>2	62	49.6
<2	30	24.0
none	24	19.2
Working Status		
Retired	55	44
Active	40	32
% Improv. L-Dopa Challenge	-	56.1 ± 0.16
Target		
STN	103	82.4
VIM	12	9.6
GPi	10	8.0
Complication		
None	83	66.4
Transient	23	18.4
Persistent	14	11.2
N.A.	5	4.0
Driving prior to DBS	77	61.6
Driving post-DBS	27	21.6
LEDD (mg)		
Preoperative	1096.0 ± 581.0	-
At last FU appointment	845.2 ± 547.6	-
% Change LEDD at last FU		−11.4 ± 58.0
FU time post DBS (months)	129.9 ± 52.8	-

PD: Parkinson’s disease; H&Y: Hoehn & Yahr scale; STN: Subthalamic Nucleus; VIM: Ventral intermediate nucleus of the Thalamus; GPi: Globus Pallidus internus; LEDD: Levodopa equivalent daily dose; FU: follow-up; DBS: deep brain stimulation; N.A.: not available.

**Table 2 jcm-12-00166-t002:** Pre- and Postoperative Neuropsychological Assessments.

	Preoperative	Postoperative	*p* Value
	mean ± Std	mean ± Std	
WAIS III (performance)	100.4 ± 15.1	98.3 ± 14.8	0.00
WAIS III (verbal)	102.1 ± 13.8	99.9 ± 14.2	0.02
WAIS III (full)	105.0 ± 13.5	102.9 ± 13.3	0.04
Memory (words)	44.2 ± 6.13	43.3 ± 6.27	n.s.
Memory (faces)	42.6 ± 5.60	41.4 ± 5.72	n.s.
Naming	21.4 ± 3.93	21.3 ± 4.04	n.s.
VOSP Letters	19.0 ± 0.98	19.1 ± 0.80	n.s.
VOSP Objects	17.4 ± 1.87	17.4 ± 1.86	n.s.
DKEFS Letters	*n* (%)	*n* (%)	n.s.
Above average	57 (45.6)	30 (24.0)	
Average	25 (20.0)	28 (22.4)	
Below Average	14 (11.2)	22 (17.6)	
N.A.	29 (23.2)	44 (35.2)	
DKEFS Category			n.s.
Above average	36 (28.8)	26 (20.8)	
Average	32 (25.6)	27 (21.6)	
Below Average	26 (20.8)	26 (20.8)	
N.A.	30 (24.0)	46 (36.8)	
Hayling Test			n.s.
Above average	4 (3.2)	10 (8.0)	
Average	69 (55.2)	50 (40.0)	
Below Average	29 (23.2)	20 (16.0)	
N.A.	23 (18.4)	45 (18.4)	
Brixton Test			n.s.
Above average	22 (17.6)	12 (9.6)	
Average	25 (20.0)	28 (22.4)	
Below Average	55 (44.0)	40 (32.0)	
N.A.	23 (28.4)	45 (36%)	

n.s.: *p* > 0.05; N.A. not available; WAIS III: Wechsler Adult Intelligent Scale part III; VOSP: Visual Object and Space Perception Battery; DKEFS: Delis-Kaplan Executive Function System.

**Table 3 jcm-12-00166-t003:** Univariate regression results of the potential epidemiological risk factors influencing driving post DBS.

	*n*	Odds Ratio	95% CI	*p* Value	R^2^
Age	85	1.170	0.488–2.808	0.725	0.002
<50 years	31				
50–69 years	52				
>70 years	2				
Profession	65	0.359	0.120–1.074	0.067	0.074
Active	31				
Retired	34				
DBS target	85	2.346	0.780–7.051	0.129	0.049
STN	69				
VIM	10				
GPi	6				
Post-op complications	81	1.354	0.541–3.391	0.517	0.008
None	64				
Transient	12				
Persistent	5				
Implant revision	85	1.442	0.272–7.661	0.667	0.003
Yes	8				
No	77				
Comorbidities	85	0.995	0.399–2.481	0.991	0.000
≤2	41				
>2	44				
Laterality at PD onset	71	1.045	0.481–2.271	0.911	0.000
Right	36				
Left	29				
PD subtype	81	2.967	1.071–8.218	0.036	0.079
Tremor-dominant	44				
Akinetic	37				
LEDD	82	0.584	0.201–1.698	0.323	0.017
Reduced	56				
Increased	26				
Driving preoperatively	79	2.567	0.522–12.626	0.246	0.028
Yes	66				
No	13				
Psychiatric pre-op assessment	71	2.153	0.681–6.807	0.191	0.036
Abnormal	24				
Unremarkable	47				
Psychiatric post-op assessment	70	3.220	1.079–9.604	0.036	0.092
Abnormal	39				
Unremarkable	31				

CI: confidence interval; DBS: deep brain stimulation; STN: Subthalamic nucleus; GPi: Globus pallidus internus; VIM: Ventral intermediate nucleus of the Thalamus; PD: Parkinson’s disease; LEDD: levodopa equivalent daily dose.

**Table 4 jcm-12-00166-t004:** Univariate regression results of the potential neuropsychological risk factors influencing driving post DBS.

	*n*	Odds Ratio	95% CI	*p* Value	R^2^
Neuropsych Pre-op	71	0.847	0.621–1.155	0.295	0.022
None	40				
Mood disorders	15				
ICB	9				
Cognitive	6				
Neuropsych Post-op	69	0.767	0.526–1.120	0.169	0.041
None	36				
Mood disorders	9				
ICB	4				
Cognitive	13				
WAIS III (performance)	57	1.111	0.369–3.346	0.851	0.001
Improved/stable	28				
Worsened	29				
WAIS III (verbal)	57	0.729	0.241–2.199	0.574	0.008
Improved/stable	27				
Worsened	30				
Memory (words)	44	0.406	0.111–1.490	0.174	0.059
Improved/stable	23				
Worsened	21				
Naming	55	1.039	0.294–3.665	0.953	0.000
Improved/stable	40				
Worsened	15				
VOSP Letters	55	2.143	0.596–7.703	0.243	0.034
Improved/stable	42				
Worsened	13				
DKEFS Letters	47	2.036	0.599–6.922	0.255	0.038
Improved/stable	26				
Worsened	21				
DKEFS Category	47	1.467	0.410–5.249	0.556	0.010
Improved/stable	32				
Worsened	15				
Hayling Test	50	9.846	1.003–96.664	0.050	0.132
Improved/stable	45				
Worsened	5				
Brixton test	51	3.556	0.692–18.282	0.129	0.063
Improved/stable	44				
Worsened	7				

CI: confidence interval; WAIS III: Wechsler Adult Intelligent Scale part III; VOSP: Visual Object and Space Perception Battery; DKEFS: Delis-Kaplan Executive Function System; ICB: impulse-control behaviours.

## Data Availability

Supporting data are available and can be provided upon request to the authors.

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
