# Peer review of "Factors Influencing Driving following DBS Surgery in Parkinson’s Disease: A Single UK Centre Experience and Review of the Literature"

_jcm, 2022, doi:10.3390/jcm12010166_

Round 1
Reviewer 1 Report (Previous Reviewer 1)
Authors seem to have addressed my comments.
Author Response
As per reviewer 1, no further actions needed.
Yours sincerely,
Reviewer 2 Report (New Reviewer)
The authors have evaluated an important parameter exploring exploring the impact of DBS on patients’ ability to drive following surgery; for which there is limited literature available.
They have thoroughly discussed the multifactorial nature rather than a direct consequence of DBS, thus highlighting the importance of routine postoperative neuropsychological evaluation in PD.Study has its importance in framing future guideline in evaluating fitness for driving .
Author Response
As per reviewer 2, no further actions needed.
Yours sincerely,
Reviewer 3 Report (New Reviewer)
The driving ability is compromised in patients with Parkinson’s disease due to motor and non-motor malfunction. The author analyzed the factor affecting driving ability in PD patients after deep drain stimulation treatment and found that the percentage of patient driving decline after the surgery. The presence of postoperative side effects was shown to be associated with driving restriction. The authors conclude that multi factors affect the driving ability after surgery rather than a direct consequence of DBS itself. This study is interesting and emphasize postoperative neuropsychological evaluation in driving ability assessment for DBS treated patient.
1, It was reported that 50% of driving-license holders still drove 12 months after DBS surgery (PMID: 26640738). However, there is a dramatic decrease of the percentage of patients who drive postoperatively. The author should explain this discrepancy.
2, Patients could benefit from DBS in driving ability with a reduction in driving errors and improvements in driving accuracy(PMID: 24353336). The author could discuss the contribution of motor improvement by DBS in driving ability.
3, DBS was reported to be be complicated by adverse neuropsychiatric side effects. (PMID: 34225945). However, the the non-motor effects were rarely observed during stimulation titration (PMID: 35029801). It would be nice if the authors discuss the incidence of side-effect in DBS.
4, The motor and non-motor effects by DBS should be discussed in driving ability assessment.
Author Response
Response to reviewers and editors
Dear Madam, dear Sir,
All the points raised by the reviewer have been addressed below and highlighted in the manuscript. Thank you very much for the invaluable contributions to this work.
- It was reported that 50% of driving-license holders still drove 12 months after DBS surgery (PMID: 26640738). However, there is a dramatic decrease of the percentage of patients who drive postoperatively. The author should explain this discrepancy.
Reply: Many thanks for your comment. Indeed, Buhmann et al (PMID: 26640738) reported a retrospective questionnaire-based survey filled out by DBS patients 4 years after surgery. In that study, ca. 50% of the active drivers reported to have resumed driving at 3-month follow-up. There are probably several reasons for the differences encountered between the cited work and our study. For instance, the studies were performed in two different countries, where driving legislations differ quite extensively. The authors reported that in only 30% of the cases the treating physician knew whether the patient was actually driving postoperatively or not, what might have been a source of bias. In some countries license holders have a legal duty to notify licensing agency of any illness that would impact on safe driving ability. Where uncertainty exists, the licensing agency may require a formal report by the treating physician before allowing the patient to resume driving. In some other countries this sort of notification is not mandatory. These aspects have been explored in the discussion (page 12).
- Patients could benefit from DBS in driving ability with a reduction in driving errors and improvements in driving accuracy(PMID: 24353336). The author could discuss the contribution of motor improvement by DBS in driving ability.
Reply: Thank you for your suggestion. This aspect has been discussed in the first paragraph of page 13.
3, DBS was reported to be be complicated by adverse neuropsychiatric side effects. (PMID: 34225945). However, the the non-motor effects were rarely observed during stimulation titration (PMID: 35029801). It would be nice if the authors discuss the incidence of side-effect in DBS.
Reply: Thank you for your suggestion. Adverse events and side-effects related to DBS have been discussed in the third paragraph of page 13.
4, The motor and non-motor effects by DBS should be discussed in driving ability assessment.
Reply: Thank you for your suggestion. This has been discussed in pages 12 and 13 of the discussion session.
Yours sincerely,

This manuscript is a resubmission of an earlier submission. The following is a list of the peer review reports and author responses from that submission.
Round 1
Reviewer 1 Report
A well designed and presented paper of a significant issue of interest in PD and social interaction arear. The MS ought to be proof read and points highlighted in the text be taken into account. Though the tables presented are informative, sometime turning these numbers to pictorial / graphical presentations help the reader get the overall message faster. I suggest doing this. I would also suggest that the notion of driving should be extended in Discussion and Conclusions to other neuropsychological concerns such as fear of falling by PD patients. overall, the paper feels a little too distant from the patient's viewpoints when it comes to the complex task of driving. As is evident to authors, freezing, movement coordination, initiation and termination etc all become a rather conscious process, cognitively, compared to non-PD individuals, especially as the disease progresses. It is important, therefore, to address how a "standardised framework" could help considering disease progression of this nature. Elaboration of this area would be helpful. In this vein, some description of how clinical decisions are currently made on driving in post-DBS PD patients would further illustrate the myriad of issues clinicians need to concern themselves with before advising their patients on driving.
Lastly, it is not know what contributions Charlotte Burford and Melika Akhbari have made in this research, as their names do not seem to appear in Author contribution section.

Author Response
Thank you very much for the helpful insights.
- Regarding a graphical representation of the data displayed in the tables, we have decided to avoid the use of figures exactly because of the extensive amount of information.
- We have addressed in more details the complexity of driving from the patient's point of view as Parkinson's disease progresses (2. Paragraph of the discussion session).
- We have also included more details concerning the decision-making process on driving following DBS surgery in PD patients (5. Paragraph of the discussion session).
- Regarding the authors' contributions, all authors were taken into consideration everywhere it says "all authors" in the session "Author Contributions".
- The manuscript has been written and revised by native English speakers.
Reviewer 2 Report
The purpose of this prospective clinical investigation was to examine neurological and cognitive markers as potential predictors of driving ability in a cohort of PD patients following DBS. Although the article is interesting, the data is poorly explored and does not indicated potential predictors associated with the ability to drive pre- and/or postoperatively.
The major concern of this manuscript is the data analysis. The authors poorly explained the statistical analysis (i.e., in some parts of the manuscript the authors mention correlation analysis, but this approach is lacking on section 2.5). More than that, the data analysis used in this manuscript is based on two group comparisons. This is surprising since the authors have access to lots of information including many different neuropsychological tests. A multivariate approach would suit better for this manuscript otherwise the conclusions are very limited. Minor concerns include the lack of Cronbach’s alpha for the neuropsychological tests.
Author Response
Thank you very much for your constructive recommendations.
We have added more information regarding the data analysis in the Methods session and included the Cronbach’s alpha value for the neuropsychological tests.
Respecting the limitations discussed in the paper, we did report factors associated with the capacity to drive following DBS surgery in this cohort. For instance, the ability to drive prior to surgery, the presence of comorbidities, the presence of an abnormal preoperative neuropsychological evaluation, the presence of any degree of cognitive impairment and the occurrence of postoperative complications were found to influence outcome. The multivariate analysis was conducted; however, it did not reveal any new relevant information apart from the data already reported in the paper.
As discussed under the light of the current literature, our data indicates that restriction to drive following DBS seems to be multifactorial rather than a direct consequence of DBS itself. There is still no consensus regarding the optimal battery of neuropsychological tests to assess patients’ fitness to drive following DBS in PD. The present paper may encourage other groups to draw attention and further discuss this important matter in the setting of larger prospective studies.
Round 2
Reviewer 2 Report
Thank you for your consideration. In the opinion of this reviewer, the topic addressed in manuscript is very important but there is no novelty. Although the authors mention that a multivariate analysis was conducted, it is still not clear how the data was analyzed.
Author Response
Thank you very much for your comments.
We have added more information regarding the data analysis in the Methods session and included the Cronbach’s alpha value for the neuropsychological tests as previously suggested.
Although several previous works have reported on the impact of ageing and of Parkinson’s disease itself on the ability to drive [1–5], only few groups have specifically investigated the influence of deep brain stimulation surgery on driving [6, 7].
We believe that our research does add novelty, considering that factors associated with the capacity to drive following DBS have been explored, reported, and discussed under the light of the literature. For instance, factors such as the ability to drive prior to surgery, the presence of comorbidities, the presence of an abnormal preoperative neuropsychological evaluation, the presence of any degree of cognitive impairment and the occurrence of postoperative complications were found to influence outcome. Therefore, our data indicates that restriction to drive following DBS seems to be multifactorial rather than a direct consequence of DBS itself.
The present paper may encourage other groups to further discuss this important matter in the setting of larger prospective studies, using both on-road drive assessments and driving simulators, in combination with neurological and neuropsychological evaluations, to better understand the potential predictors of postoperative safe driving. This will potentially provide a framework which patients, medical staff and driving authorities will be able to refer to in evaluating fitness to drive following DBS in PD.
References
- Buhmann C, Vesper J, Oelsner H (2018) [Driving ability in Parkinson’s disease]. Fortschr Neurol Psychiatr 86:43–48. https://doi.org/10.1055/s-0043-110051
- Dubinsky RM, Gray C, Husted D, et al (1991) Driving in Parkinson’s disease. Neurology 41:517–520. https://doi.org/10.1212/wnl.41.4.517
- Classen S, Witter DP, Lanford DN, et al (2011) Usefulness of screening tools for predicting driving performance in people with Parkinson’s disease. Am J Occup Ther 65:579–588. https://doi.org/10.5014/ajot.2011.001073
- Meindorfner C, Körner Y, Möller JC, et al (2005) Driving in Parkinson’s disease: mobility, accidents, and sudden onset of sleep at the wheel. Mov Disord 20:832–842. https://doi.org/10.1002/mds.20412
- Crizzle AM, Classen S, Uc EY (2012) Parkinson disease and driving: an evidence-based review. Neurology 79:2067–2074. https://doi.org/10.1212/WNL.0b013e3182749e95
- Buhmann C, Maintz L, Hierling J, et al (2014) Effect of subthalamic nucleus deep brain stimulation on driving in Parkinson disease. Neurology 82:32–40. https://doi.org/10.1212/01.wnl.0000438223.17976.fb
- Charmley AR, Kimber T, Mahant N, Lehn A (2021) Driving restrictions following deep brain stimulation surgery. BMJ Neurol Open 3:e000210. https://doi.org/10.1136/bmjno-2021-000210
